# Uncovering Directions of Instability via Quadratic Approximation of Deep Neural Loss in Reinforcement Learning

## Abstract

Learning in MDPs with highly complex state representations is currently possible due to multiple advancements in reinforcement learning algorithm design. However, this incline in complexity, and furthermore the increase in the dimensions of the observation came at the cost of non-robustness that can be taken advantage of (i.e. moving along worst-case directions in the observation space). To solve this policy instability problem we propose a novel method to ascertain the presence of these non-robust directions via quadratic approximation of the deep neural policy loss. Our method provides a theoretical basis for the fundamental cut-off between stable observations and non-robust observations. Furthermore, our technique is computationally efficient, and does not depend on the methods used to produce the worst-case directions. We conduct extensive experiments in the Arcade Learning Environment with several different non-robust alteration techniques. Most significantly, we demonstrate the effectiveness of our approach even in the setting where alterations are explicitly optimized to circumvent our proposed method.

## 1 Introduction

Since Mnih et al. (2015) showed that deep neural networks can be used to parameterize reinforcement learning policies, there has been substantial growth in new algorithms and applications for deep reinforcement learning. While this progress has resulted in a variety of new capabilities for reinforcement learning agents, it has at the same time introduced new challenges due to the non-robustness of deep neural networks to imperceptible adversarial perturbations originally discovered by Szegedy et al. (2014). In particular, Huang et al. (2017); Kos & Song (2017) showed that the non-robustness of neural networks to adversarial perturbations extends to the deep reinforcement learning domain, where applications such as autonomous driving, automatic financial trading or healthcare decision making cannot tolerate such a vulnerability.

There has been a significant amount of effort in trying to make deep neural networks robust to adversarial perturbations (Goodfellow et al., 2015; Madry et al., 2018; Pinto et al., 2017). However, in this arms race it has been shown that deep reinforcement learning policies learn adversarial features independent from their worst-case (i.e. adversarial) training techniques (Korkmaz, 2022). More intriguingly, a line of work has focused on showing the inevitability of adversarial examples and the intrinsic difficulty of learning robust models (Dohmatob, 2019; Mahloujifar et al., 2019; Gourdeau et al., 2019). Given that it may not be possible to make DNNs completely robust to adversarial examples, a natural objective is to instead attempt to detect the presence of adversarial manipulations.

In this paper we propose a novel identification method for adversarial directions in the deep neural policy manifold. Our study is the first one that focuses on detection of non-robust directions in the deep reinforcement learning neural loss landscape. Our approach relies on differences in the curvature of the neural policy in the neighborhood of an adversarial direction when compared to a baseline state observation. At a high level our method is based on the intuition that while baseline states have neighborhoods determined by an optimization procedure intended to learn a policy that works well across all states, each non-robust direction is the output of some local optimization in the neighborhood of one particular state. Our proposed method is computationally efficient, requiring only one gradient computation and two policy evaluations, requires no training that depends on the

method used to compute the adversarial direction, and is theoretically well-founded. Hence, our study focuses on identification of non-robust directions and makes the following contributions:

- Our paper is the first to focus on identification of adversarial directions in the deep reinforcement learning policy manifold.
- We propose a novel method, Identification of Non-Robust Directions (INRD), to detect adversarial state manipulations based on the local curvature of the neural network policy. INRD is independent of the method used to generate the adversarial direction, computationally efficient, and theoretically justified.
- We conduct experiments in various MDPs from the Arcade Learning Environment that demonstrate the effectiveness of INRD in identifying adversarial directions computed via several state-of-the-art adversarial attack methods.
- Most importantly, we demonstrate that INRD remains effective even against multiple methods for generating non-robust directions specifically designed to evade INRD.

## 2 RELATED WORK AND BACKGROUND

**Deep Reinforcement Learning:** In this paper we focus on discrete action set Markov Decision Processes (MDPs) which are given by a continuous set of states $\mathbb{S}$, a discrete set of actions $\mathbb{A}$, a transition probability function $P : \mathbb{S} \times \mathbb{A} \times \mathbb{S} \to \mathbb{R}$, and a reward function $\mathcal{R} : \mathbb{S} \times \mathbb{A} \times \mathbb{S} \to \mathbb{R}$. A policy $\pi : \mathbb{S} \to \mathcal{P}(\mathbb{A})$ assigns a probability distribution on actions $\pi(\cdot|s)$ to each state $s$. The goal in reinforcement learning is to learn the state-action value function that maximizes expected cumulative discounted rewards $R = \mathbb{E}_{a_t \sim \pi(s_t, \cdot)} \sum_t \gamma^t \mathcal{R}(s_t, a_t, s_{t+1})$ by taking action $a$ in state $s$. The temporal difference learning is achieved by one step Q-learning which updates $Q(s_t, a_t)$ by

$$Q(s_t, a_t) + \alpha[\mathcal{R}_{t+1} + \gamma \max_a Q(s_{t+1}, a) - Q(s_t, a_t)].$$

**Adversarial Examples:** Goodfellow et al. (2015) introduced the fast gradient method (FGM) for producing adversarial examples for image classification. The method is based on taking the gradient of the training cost function $J(x, y)$ with respect to the input image, and bounding the perturbation by $\epsilon$ where $x$ is the input image and $y$ is the output label. Later, an iterative version of FGM called I-FGM was proposed by Kurakin et al. (2016). This is also often referred to as Projected Gradient Descent (PGD) as in (Madry et al., 2018) where the I-FGM update is

$$x_{\text{adv}}^{N+1} = \text{clip}_\epsilon(x_{\text{adv}}^N + \alpha \text{sign}(\nabla_x J(x_{\text{adv}}^N, y))). \tag{1}$$

where $x_{\text{adv}}^0 = x$. Dong et al. (2018) further modified I-FGM by introducing a momentum term in the update, yielding a method called MI-FGSM. Korkmaz (2020) later proposed a Nesterov-momentum based approach for the deep reinforcement learning domain. The DeepFool method of Moosavi-Dezfooli et al. (2016) is an alternative approach to those based on FGSM. DeepFool performs iterative projection to the closest separating hyperplane between classes. Another alternative approach proposed by Carlini & Wagner (2017a) is based on finding a minimal perturbation that achieves a different target class label. The approach is based on minimizing the loss

$$\min_{s^{\text{adv}} \in \mathbb{S}} c \cdot J(s^{\text{adv}}) + \left\| s^{\text{adv}} - s \right\|_2^2 \tag{2}$$

where $s$ is the clean input, $s_{\text{adv}}$ is the adversarial example, and $J(s)$ is a modified version of the cost function used to train the network. Chen et al. (2018) proposed a variant of the Carlini & Wagner (2017a) formulation that adds an $\ell_1$-regularization term to produce sparser adversarial examples,

$$\min_{s^{\text{adv}} \in \mathbb{S}} c \cdot J(s^{\text{adv}}) + \lambda_1 \left\| s^{\text{adv}} - s \right\|_1 + \lambda_2 \left\| s^{\text{adv}} - s \right\|_2^2 \tag{3}$$

Our method focusing on identifying non-robust directions in the deep neural policy manifold is the first method to investigate detection of adversarial manipulations in deep reinforcement learning. Our identification method does not require modifying the training of the neural network, does not require any training tailored to the adversarial method used, and uses only two neural network function evaluations and one gradient computation.

**Adversarial Deep Reinforcement Learning:** The adversarial problem initially has been investigated by Huang et al. (2017) and Kos & Song (2017) concurrently. In this work the authors show that perturbations computed via FGSM result in extreme performance loss on the learnt policy. Lin et al. (2017) and Sun et al. (2020) focused on timing strategies in the adversarial formulation and utilized the Carlini & Wagner (2017a) method to produce the perturbations. While there is a reasonable body of work focused on finding efficient and effective adversarial perturbations, a substantial body of work focused on building agents robust to these perturbations. Mandlekar et al. (2017) proposed to utilize FGSM perturbations during training time to obtain more robust agents. Pinto et al. (2017) modeled the adversarial interaction as a zero sum game and proposed a joint training strategy to increase robustness in the continuous action space setting. Recently, Gleave et al. (2020) considered an adversary who is allowed to take natural actions in a given environment instead of $\ell_p$-norm bounded perturbations and modeled the adversarial relationship as a zero sum Markov game. However, recent concerns have been raised on the robustness of adversarial training methods by Korkmaz (2022). In this paper the authors show that the state-of-the-art adversarial training techniques end up learning similar non-robust features. Thus, with the rising concerns on robustness of recent proposed adversarial training techniques our work aims to solve the adversarial problem from a different perspective by detecting adversarial directions.

## 3 IDENTIFICATION OF NON-ROBUST DIRECTIONS (INRD)

In this section we give the high-level motivation for and formal description of our identification method. We begin by introducing necessary notation and definitions. We denote an original clean state by $\bar{s}$ and an adversarially perturbed state by $s^{\text{adv}}$.

**Definition 3.1.** The *cost of a state*, $J(s, \tau)$, is defined as the cross entropy loss between the policy $\pi(a|s)$ of the agent, and a target distribution on actions $\tau(a)$.

$$J(s, \tau) = -\sum_a \tau(a) \log(\pi(a|s)) \tag{4}$$

**Definition 3.2.** The *argmax policy*, $\pi^*(a|s)$, is defined as the distribution which puts all probability mass on the highest weight action of $\pi(a|s)$.

$$\pi^*(a|s) = \mathbb{1}(a = \arg\max_{a'} \pi(a'|s)) \tag{5}$$

We use the following notation for the gradient and Hessian with respect to states $s$:

$$\nabla_s J(s_0, \tau_0) = \nabla_s J(s, \tau)|_{s=s_0, \tau=\tau_0}$$
$$\nabla_s^2 J(s_0, \tau_0) = \nabla_s^2 J(s, \tau)|_{s=s_0, \tau=\tau_0}$$

### 3.1 FIRST-ORDER IDENTIFICATION OF NON-ROBUST DIRECTIONS (FO-INRD)

As a naive baseline we first describe an identification method based on estimating how much the cost function $J(s, \tau)$ varies under small perturbations. Prior work of Roth et al. (2019); Hu et al. (2019) has shown that the behavior of deep neural network classifiers under small, random perturbations is different at clean versus adversarial examples. Therefore, a natural baseline detection method is: given an input state $s_0$ sample a small random perturbation $\eta \sim \mathcal{N}(0, \epsilon I)$ and compute,

$$\mathcal{K}(s_0, \eta) = J(s_0 + \eta, \pi^*(\cdot|s_0)) - J(s_0, \pi^*(\cdot|s_0)). \tag{6}$$

The first-order identification method proceeds by first estimating the mean and the variance of $\mathcal{K}$ over a clean run of the agent in the environment. Next a threshold $t$ is chosen so that a desired false positive rate (FPR) is achieved (i.e. some desired fraction of the states in the clean run lie more than $t$ standard deviations from the mean). Finally, at test time a state encountered by the agent is classified as adversarial if it is at least $t$ standard deviations away from the mean. Otherwise the state is classified as clean. As a first attempt, the first-order method can be naturally interpreted as a finite-difference approximation to the magnitude of the gradient at $s_0$. If we assume that the first-order Taylor approximation of $J$ is accurate in a ball of radius $r > \epsilon$ centered at $s_0$, then

$$J(s_0 + \eta, \pi^*(\cdot|s_0)) \approx J(s_0, \pi^*(\cdot|s_0)) + \nabla_s J(s_0, \pi^*(\cdot|s_0)) \cdot \eta.$$

Therefore,

$$\mathcal{K}(s_0, \eta) \approx \nabla_s J(s_0, \pi^*(\cdot|s_0)) \cdot \eta. \tag{7}$$

Thus, for $\eta \sim \mathcal{N}(0, \epsilon I)$ the test statistic $\mathcal{K}(s_0, \eta)$ is approximately distributed as a Gaussian with mean 0 and variance $\epsilon^2 \|\nabla_s J(s_0, \pi^*(\cdot|s_0))\|^2$. Under this interpretation one would expect the test statistics for clean and adversarial states to have the same mean with potentially different standard deviations, possibly making it hard to distinguish clean from adversarial. However, this is not what we observe empirically, and in fact the first-order method does a decent job of detecting adversarial examples. The method works because, in fact, the mean of $\mathcal{K}(\bar{s}, \eta)$ for clean examples $\bar{s}$ is reasonably well separated from the mean of $\mathcal{K}(s^{\text{adv}}, \eta)$ for adversarial examples $s^{\text{adv}}$. The empirical performance of the first-order method thus indicates that the assumption of accuracy for the first-order Taylor approximation of $J$ does not hold in practice. This leads naturally to the consideration of information on the second derivatives (i.e. the local quadratic approximation) of $J$ in order to identify non-robust directions.

## 3.2 SECOND-ORDER IDENTIFICATION OF NON-ROBUST DIRECTIONS (SO-INRD)

The second-order identification method is based on measuring the local curvature of the cost function $J(s, \tau)$. The method exploits the fact that $J(s, \tau)$ will have larger negative curvature at a clean example as compared to an adversarial example. In particular, the high level theoretical motivation for this approach is that adversarial examples are the output of a local optimization procedure which attempts to find a nearby perturbed state $s^{\text{adv}}$ with a low value for the cost $J(s^{\text{adv}}, \tau)$ for some $\tau \neq \pi^*(\cdot|\bar{s})$. A direction of large negative curvature for $J(s^{\text{adv}}, \tau)$ indicates that a very small perturbation along this direction could dramatically decrease the cost function. Therefore, such points are likely to be unstable for local optimization procedures attempting to minimize the cost function in a small neighborhood. On the other hand, the curvature of $J(s, \tau)$ at a clean state $\bar{s}$ is determined by the overall algorithm used to train the deep reinforcement learning agent. This algorithm optimizes the parameters of the neural network policy while considering all states visited during training, and thus is not likely to be heavily overfit to the state $\bar{s}$. In particular, we expect larger negative curvature at $\bar{s}$ than at an adversarial example $s^{\text{adv}}$. We make the connection between negative curvature and instability for local optimization formal in Section 3.3. Based on the above discussion, a natural choice of metric for distinguishing adversarial versus clean examples is the most negative eigenvalue of the Hessian $\lambda_{\min}\left(\nabla_s^2 J(s_0, \pi^*(\cdot|s_0))\right)$. While this is the most natural measurement of curvature, it requires computing the eigenvalues of a matrix whose number of entries are quadratic in the input dimension. Since the input is very high-dimensional, and we would like to perform this computation in real-time for every state visited by the agent, computing the value $\lambda_{\min}$ is computationally prohibitive. Instead we approximate this value by measuring the curvature along a direction which is correlated with the negative eigenvectors of the Hessian. Given this direction, the value that we measure is the accuracy of the first order Taylor approximation of the cost of the given state $J(s, \tau)$. We denote the first order Taylor approximation at the state $s_0$ in direction $\eta$ by

$$\tilde{J}(s_0, \eta) = J(s_0, \pi^*(\cdot|s_0)) + \nabla_s J(s_0, \pi^*(\cdot|s_0)) \cdot \eta.$$

The metric we will use to detect adversarial examples is the finite-difference approximation

$$\mathcal{L}(s_0, \eta) = J(s_0 + \eta, \pi^*(\cdot|s_0)) - \tilde{J}(s_0, \eta). \tag{8}$$

To see formally that Equation (8) gives an approximation of the most negative eigenvector of the Hessian, we will assume that the cost function $J(s, \tau)$ is well approximated by its quadratic Taylor approximation at the point $s_0$ i.e.

$$J(s_0 + \eta, \pi^*(\cdot|s_0)) \approx J(s_0, \pi^*(\cdot|s_0)) + \nabla_s J(s_0, \pi^*(\cdot|s_0)) \cdot \eta + \eta^\top \nabla_s^2 J(s_0, \pi^*(\cdot|s_0))\eta \tag{9}$$

for a small enough perturbation $\eta$. Substituting the above formula into Equation (8) yields

$$\mathcal{L}(s_0, \eta) \approx \eta^\top \nabla_s^2 J(s_0, \pi^*(\cdot|s_0))\eta \tag{10}$$

The above quadratic form is minimized when $\eta$ lies in the same direction as the most negative eigenvector of the Hessian, in which case

$$\mathcal{L}(s_0, \eta) \approx \lambda_{\min}\left(\nabla_s^2 J(s_0, \pi^*(\cdot|s_0))\right) \|\eta\|_2^2 \tag{11}$$

We choose the sign of the gradient direction for measuring the accuracy of the first order Taylor approximation. To motivate this choice note that $-\nabla_s J(s, \tau)$ is locally the direction of steepest

---

**Algorithm 1:** Second Order Identification of Non-Robust Directions (SO-INRD)

---

**Input:** The clean run mean $\bar{\mathcal{L}}$ and variance $\sigma^2(\mathcal{L})$, identification threshold $t > 0$, parameter $\epsilon > 0$.

**for** states $s_i$ visited by deep reinforcement learning policy **do**

$\quad \eta_i = \epsilon \dfrac{\text{sign}(\nabla_s J(s_i, \pi^*(\cdot|s_i)))}{\|\nabla_s J(s_i, \pi^*(\cdot|s_i))\|_2}$

$\quad \tilde{J}(s_i, \eta_i) = J(s_i, \pi^*(\cdot|s_i)) + \nabla_s J(s_i, \pi^*(\cdot|s_i)) \cdot \eta_i$

$\quad \mathcal{L}(s_i, \eta_i) = J(s_i + \eta_i, \pi^*(\cdot|s_i)) - \tilde{J}(s_i, \eta_i)$

$\quad$ **if** $|\mathcal{L}(s_i, \eta_i) - \bar{\mathcal{L}}| > t \cdot \sigma(\mathcal{L})$ **then**

$\quad \quad$ Label state $s_i$ as a non-robust direction

$\quad$ **end if**

**end for**

---

decrease for the cost function. If the gradient direction additionally has negative curvature of large magnitude, then small perturbations along this direction will result in even more rapid decrease in the cost function value than predicted by the first-order gradient approximation. Note that this can be true even if the gradient itself has small magnitude, as long as the negative curvature is large enough. Thus, by the discussion at the beginning of Section 3.2, adversarial examples are likely to have relatively smaller magnitude negative curvature in the gradient direction than clean examples. Formally, for $\epsilon > 0$ we set

$$\eta(s_0) = \epsilon \frac{\text{sign}\left(\nabla_s J(s_0, \pi^*(\cdot|s_0))\right)}{\|\nabla_s J(s_0, \pi^*(\cdot|s_0))\|_2}. \tag{12}$$

To calibrate the detection method we record the mean $\bar{\mathcal{L}} = \mathbb{E}_s[\mathcal{L}(s, \eta(s))]$ and variance $\sigma^2(\mathcal{L}) = \text{Var}_s[\mathcal{L}(s, \eta(s))]$ of our proposed test statistic over states from a clean run of the policy in the MDP. Then at test time we set a threshold $t > 0$, and for each state $s_i$ visited by the agent test if

$$|\mathcal{L}(s_i, \eta(s_i)) - \bar{\mathcal{L}}| > t\sigma(\mathcal{L}). \tag{13}$$

If the threshold of $t$ standard deviations is exceeded we classify the state $s_i$ as adversarial, and otherwise classify it as clean. Pseudo-code for the second order method is given in Algorithm 1.

## 3.3 Negative Curvature and Instability of Local Optimization

In this section we formalize the connection between negative curvature and instability for local optimization procedures that motivated our definition of $\mathcal{L}(s, \eta)$. Given a state $s_0$ and a target distribution $\tau \neq \pi^*(\cdot|s_0)$, we assume the adversary is trying to find a state $s^{\text{adv}}$ minimizing $J(s^{\text{adv}}, \tau)$ among all states close to $s_0$ by some metric. Formally, let $D_{s_0}(s) \geq 0$ be a convex function of $s$ that should be thought of as measuring distance to $s_0$. One standard choice for the distance function is $D_{s_0}(s) = \|s - s_0\|_p^p$. We model the adversary as minimizing the loss

$$f(s) = J(s, \tau) + D_{s_0}(s). \tag{14}$$

In particular, we make the following assumption:

**Assumption 3.1.** *The adversarial state $s^{\text{adv}}$ is a local minimum of $f(s)$.*

Of course this assumption is violated in practice since different methods used to compute adversarial directions optimize objective functions other than $f$, and do not necessarily always converge to a local minimum. Nevertheless the assumption allows us to make formal qualitative predictions about the behavior of the second-order identification method that correspond well with empirical results across a broad variety of methods for generating adversarial directions. We now state our main result lower bounding the curvature of $J(s^{\text{adv}}, \tau)$.

**Proposition 3.2.** *For $c > 0$ assume that the maximum eigenvalue of the Hessian $\nabla_s^2 D_{s_0}(s)$ is bounded by $c$. If $s^*$ is a local minimum of $f(s)$ then $\lambda_{min}(\nabla_s^2 J(s^*, \tau)) \geq -c$*

*Proof.* Let $v$ be the eigenvector of $\nabla_s^2 J(s^*, \tau)$ corresponding to the minimum eigenvalue. At a local minimum $s^*$ of $f(s)$ the Hessian $\nabla_s^2 f(s^*)$ must be positive semi-definite. Therefore,

$$0 \leq v^\top \nabla_s^2 f(s^*) v = v^\top \nabla_s^2 J(s^*, \tau) v + v^\top \nabla_s^2 D_{s_0}(s^*) v$$
$$\leq \lambda_{\min}(\nabla_s^2 J(s^*, \tau)) + c$$

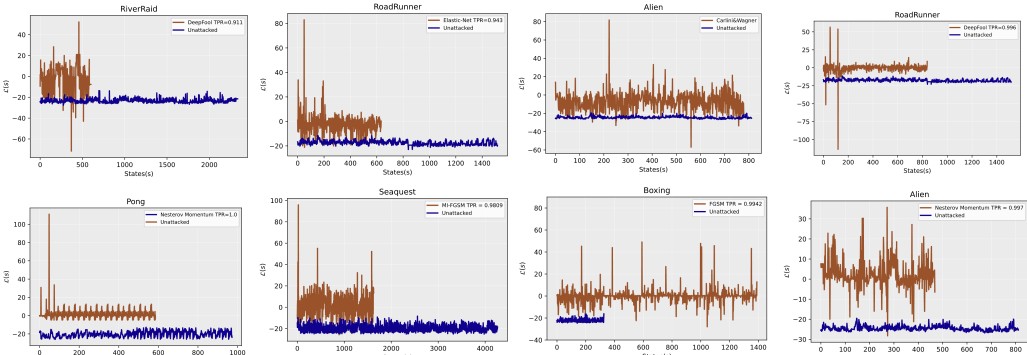

Figure 1: $\mathcal{L}(s)$ for our proposed method SO-INRD vs visited states with corresponding TPR values for the following attack methods: FGSM, MI-FGSM, Nesterov, DeepFool, Carlini&Wagner, Elastic Net Method. TPR values shown in the upper right box of the figure when FPR is equal to 0.01.

Rearranging the above inequality completes the proof. □

The second order conditions for a local minimum of $f$ imply a lower bound on the smallest eigenvalue of $\nabla_s^2 J(s^*, \tau)$. Thus, by Assumption 3.1, we obtain a lower bound on $\lambda_{\min}(\nabla_s^2 J(s^{\mathrm{adv}}, \tau))$. The assumption that the maximum eigenvalue of the Hessian $\nabla_s^2 D_{s_0}(s)$ is bounded by $c$ is satisfied for example when $D_{s_0}(s) = \frac{c}{2}\|s - s_0\|_2^2$. In contrast, the local curvature of the cost function $J(s, \tau)$ at a clean example is determined by an optimization procedure that updates the *weights* $\theta$ of the neural network policy rather than the states $s$. If we write $J_\theta(s, \tau)$ to make explicit the dependence on the weights, then the second order conditions for optimizing the original neural network apply to the Hessian with respect to weights $\nabla_\theta^2 J_\theta(s, \tau)$ rather than the Hessian with respect to states $\nabla_s^2 J_\theta(s, \tau)$. Additionally, first order optimality conditions can help to justify the choice of $\nabla_s J(s, \tau)$ as a good direction to check for negative curvature. Indeed by the first order conditions, at a local optimum $s^*$ of $f(s)$ we have

$$0 = \nabla_s f(s^*) = \nabla_s J(s^*, \tau) + \nabla_s D_{s_0}(s^*). \tag{15}$$

Therefore, $\nabla_s J(s^*, \tau) = -\nabla_s D_{s_0}(s^*)$. So assuming the adversary finds a local optimum, $\nabla_s J(s, \tau)$ points in a direction that decreases the distance function $D_{s_0}(s^*)$. Thus sufficiently negative curvature in the direction of $\nabla_s J(s, \tau)$ implies not only that $s$ is not a local minimum of $f$, but also that the distance function $D_{s_0}(s)$ can be decreased by moving along this direction of negative curvature. To summarize, we have shown that second order optimality conditions arising from computing an adversarial example give rise to lower bounds on the smallest eigenvalue of the Hessian $\lambda_{\min}(\nabla_s^2 J(s, \tau))$. The function $\mathcal{L}(s, \eta)$ used to identify adversarial directions for SO-INRD is a finite difference approximation to

$$\eta^\top \nabla_s^2 J(s, \tau)\eta \geq \lambda_{\min}(\nabla_s^2 J(s, \tau))\|\eta\|^2.$$

Therefore the results of this section imply that $\mathcal{L}(s, \eta)$ should be larger at adversarial examples than clean examples.

## 4 EXPERIMENTS

In our experiments agents are trained with DDQN Wang et al. (2016) in the Arcade Learning Environment (ALE) Bellemare et al. (2013) from OpenAI Brockman et al. (2016). For a baseline we compare FO-INRD and SO-INRD with the detection method of OAO proposed by Roth et al. (2019), which is based on estimating the average change in the odds ratio between classes under noise.

In Figure 1 we plot the value of $\mathcal{L}(s)$ over states for various games without an adversarial attack and under adversarial attack with the following methods: Carlini & Wagner, Elastic Net, Nesterov Momentum, DeepFool, MIFGSM and FGSM. We show in the legends of Figure 1 the true positive rate (TPR) values for the different attacks when false positive rate (FPR) is equal to 0.01. The value of $\mathcal{L}(s)$ for clean states is generally well-concentrated and negative. On the other hand, for states computed by the different adversarial attack methods $\mathcal{L}(s)$ is clearly larger, matching the predictions

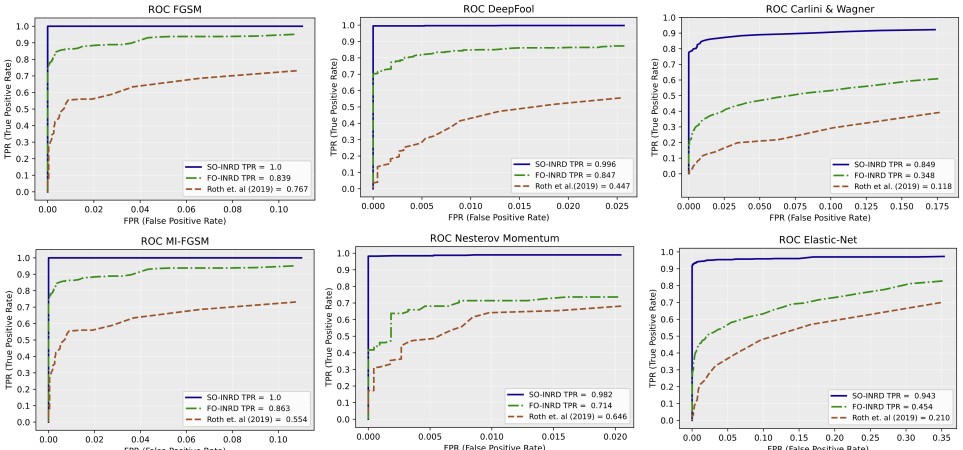

Figure 2: ROC curves of FO-INRD, SO-INRD and OAO method for the following attack methods: FGSM, MI-FGSM, Nesterov Momentum, DeepFool, Carlini&Wagner, Elastic Net Method in RoadRunner. TPR values shown in the lower right box of the figure when FPR is equal to 0.01.

Table 1: True Positive Rates (TPR) for FGSM, MI-FGSM, Nesterov Momentum, Carlini&Wagner, Elastic-Net and DeepFool when False Positive Rate (FPR) is equal to 0.01. The proposed methods SO-INRD and FO-INRD are evaluated, and compared with Roth et al. (OAO) in Riverraid, RoadRunner, Alien, Seaquest, Boxing, Pong, and Robotank games. More results for different FPR values are reported in the supplementary material.

| Detection Method-Attack Method | RiverRaid | RoadRunner | Alien | Seaquest | Boxing | Pong | Robotank |
|---|---|---|---|---|---|---|---|
| SO-INRD FGSM | 0.997 | 1.0 | 1.0 | 0.995 | 0.994 | 1.0 | 0.999 |
| FO-INRD FGSM | 0.990 | 0.843 | 0.803 | 0.931 | 0.793 | 0.622 | 0.413 |
| OAO FGSM | 0.681 | 0.767 | 0.885 | 0.403 | 0.264 | 0.424 | 0.911 |
| SO-INRD M-IFGSM | 0.998 | 1.0 | 1.0 | 0.985 | 0.910 | 1.0 | 0.985 |
| FO-INRD M-IFGSM | 0.952 | 0.863 | 0.991 | 0.981 | 0.827 | 0.622 | 0.470 |
| OAO M-IFGSM | 0.775 | 0.554 | 0.929 | 0.581 | 0.499 | 0.679 | 0.777 |
| SO-INRD Nesterov Momentum | 0.995 | 0.989 | 0.996 | 0.952 | 0.865 | 1.0 | 0.954 |
| FO-INRD Nesterov Momentum | 0.990 | 0.714 | 0.997 | 0.979 | 0.746 | 0.633 | 0.574 |
| OAO Nesterov Momentum | 0.785 | 0.646 | 0.925 | 0.671 | 0.517 | 0.687 | 0.753 |
| SO-INRD Carlini&Wagner | 0.910 | 0.988 | 0.945 | 0.723 | 0.856 | 0.850 | 0.713 |
| FO-INRD Carlini&Wagner | 0.695 | 0.594 | 0.642 | 0.516 | 0.785 | 0.494 | 0.119 |
| OAO Carlini&Wagner | 0.036 | 0.118 | 0.018 | 0.004 | 0.016 | 0.028 | 0.032 |
| SO-INRD Elastic Net | 0.777 | 0.943 | 0.875 | 0.687 | 0.770 | 0.736 | 0.815 |
| FO-INRD Elastic Net | 0.685 | 0.454 | 0.561 | 0.502 | 0.743 | 0.361 | 0.212 |
| OAO Elastic Net | 0.124 | 0.210 | 0.060 | 0.014 | 0.150 | 0.092 | 0.056 |
| SO-INRD DeepFool | 0.914 | 0.996 | 0.993 | 0.860 | 0.951 | 0.889 | 0.900 |
| FO-INRD DeepFool | 0.841 | 0.847 | 0.936 | 0.777 | 0.928 | 0.796 | 0.268 |
| OAO DeepFool | 0.397 | 0.447 | 0.611 | 0.234 | 0.381 | 0.367 | 0.607 |

of Proposition 3.2. The fact that $\mathcal{L}(s)$ is consistently larger at adversarial examples across a wide variety of adversarial perturbation methods indicates that Assumption 3.1 qualitatively captures the behavior of these methods. In particular the FGSM-based methods and DeepFool do not explicitly optimize an objective function of the form $f(s) = J(s, \tau) + D_{s_0}(s)$ as in Assumption 3.1. However, by enforcing a constraint on the distance of the adversarial example from the original clean example, these methods implicitly solve an optimization problem of the form given in (14), and thus exhibit the qualitative behavior predicted by Proposition 3.2.

In Table 1 we show TPR values for FO-INRD, SO-INRD, and the OAO method under the FGSM, MI-FGSM, Nesterov Momentum, DeepFool, Carlini&Wagner, and Elastic-Net attacks when FPR is equal to 0.01. For all of the attack methods in all of the environments SO-INRD is able to detect

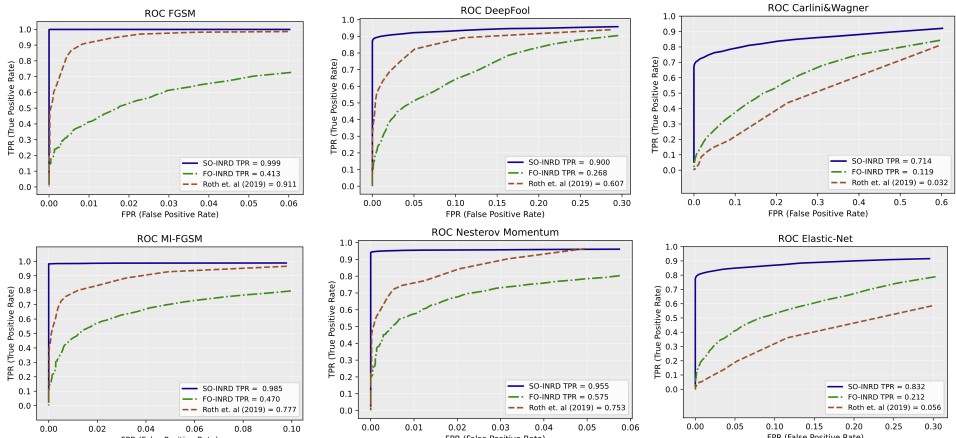

Figure 3: ROC curves of FO-INRD, SO-INRD and OAO method for the following attack methods: FGSM, MI-FGSM, Nesterov Momentum, DeepFool, Carlini&Wagner, Elastic Net Method in Robotank. TPR values are reported in the lower right box of the figure when FPR is equal to $0.01$.

Table 2: TPR for Feature Matching for SO-INRD and OAO method FPR=0.01

| Feature Matching | Riverraid | RoadRunner | Alien | Seaquest | Boxing | Robotank |
|---|---|---|---|---|---|---|
| SO-INRD | 0.882 | 0.863 | 0.9016 | 0.955 | 0.988 | 0.8978 |
| OAO Method | 0.0088 | 0.006 | 0.007 | 0.0146 | 0.0106 | 0.0158 |

adversarial perturbations with large TPR. SO-INRD outperforms the other detection methods in all cases except for Nesterov Momentum in Alien and Seaquest where FO-INRD has TPR 0.997 and 0.980 while SO-INRD has 0.996 and 0.952. We also observe that while the perturbations computed by FGSM, MI-FGSM, Nesterov Momentum can generally be detected with large TPR values by all the detection methods, the perturbations computed by Carlini&Wagner and the Elastic-Net method are more difficult to detect. Despite the difficulty, SO-INRD achieves TPR values ranging from 0.713 to 0.988 for Carlini&Wagner, and TPR values ranging from 0.687 to 0.943 for Elastic-Net when FPR is equal to 0.01. In Figure 2 and Figure 3 we show ROC curves for each detection method under the FGSM, MI-FGSM, Nesterov Momentum, DeepFool, Carlini&Wagner and Elastic-Net method for RoadRunner and Robotank respectively. In Robotank the OAO method outperforms FO-INRD and even approaches the TPR of SO-INRD for high FPR under FGSM, MI-FGSM, Nesterov Momentum and DeepFool. However for the Carlini&Wagner and Elastic-Net attacks, SO-INRD has a much higher TPR across a wide range of FPR levels.

## 5 COMPUTING ADVERSARIAL DIRECTIONS SPECIFICALLY TO EVADE INRD

Recently, Tramer et al. (2020) introduced a comprehensive methodology for tailoring the optimization procedure used to produce adversarial examples in order to overcome detection and defense methods. In particular, the high level idea is to keep the attack as simple as possible while still accurately targeting the detection method. More specifically, the methodology is based on designing an attack based on gradient descent on some loss function. Further, minimizing the loss function should correspond closely to subverting the full detection method while still being possible to optimize. Critically, the authors highlight that while the choice of loss function to optimize can be a difficult task, the use of "feature matching" Gowal et al. (2019) can circumvent most of the current detection methods. We now describe how we applied the methodology discussed above to design detection aware adversaries for SO-INRD. As a first attempt, we tested the "feature matching" approach that was used to break the OAO detection method in Tramer et al. (2020). This approach attempts to match the logits of the adversarial example to those of a clean example from a different class in order to evade detection. To optimize the loss for this method we used up to 1000 PGD iterations, and we searched step size varying from 0.01 to $10^{-6}$. We find that this method succeeds in reducing the TPR of the OAO method to nearly zero. It is also able to slightly reduce the TPR of our SO-INRD method

Table 3: TPR values of INRD in the presence of a identification aware adversary when FPR=0.01.

| Detection Method | RiverRaid | RoadRunner | Alien | Seaquest | Boxing | Pong | Robotank |
|---|---|---|---|---|---|---|---|
| SO-INRD — C&W | 0.650 | 0.849 | 0.445 | 0.381 | 0.710 | 0.712 | 0.657 |
| FO-INRD — C&W | 0.346 | 0.348 | 0.351 | 0.193 | 0.621 | 0.325 | 0.0973 |

(see results in Table 2). However, as we will see next, a larger reduction in the TPR of SO-INRD can be achieved by optimizing a modified version of the loss from Carlini & Wagner (2017b). Our next attempt is based on a modification of the Carlini & Wagner (2017b) formulation to additionally minimize the cost function $\mathcal{L}(s)$ used in SO-INRD,

$$\min_{s^{\mathrm{adv}} \in \mathbb{S}} c \cdot J(s^{\mathrm{adv}}) + \left\| s^{\mathrm{adv}} - s \right\|_2^2 + \lambda \cdot \mathcal{L}(s^{\mathrm{adv}}). \tag{16}$$

Recall that $\mathcal{L}(s)$ is consistently larger at adversarial examples than at clean examples. Thus the above optimization problem attempts to find adversarial examples with as small values of $\mathcal{L}(s)$ as possible. Since the function $\mathcal{L}(s)$ involves taking the sign of the gradient, we use Backwards Pass Differentiable Approximation (BPDA) as introduced in Carlini & Wagner (2017b) to compute the gradients. However, we also tried designing an adversary with a fully differentiable cost function by using a perturbation in the gradient direction (without the sign). We found that this fully differentiable adversary performed significantly worse than the one based on BPDA. We conducted exhaustive grid search over all the parameters in this optimization method: learning rate, iteration number, confidence parameter $\kappa$, and objective function parameter $\lambda$. In C&W we used up to 30000 iterations to find adversarial examples to bypass detection methods. We searched the confidence parameter from 0 to 50, the learning rate from 0.001 to 0.1, and $\lambda$ from 0.001 to 10. In our grid search over these hyperparameters we found that there is a trade-off between the attack success rate and the detection of the perturbations. In other words, if we optimize the perturbation to be undetectable the success rate of the perturbation (i.e. the rate at which the perturbation actually makes the agent choose a non-optimal action) decreases. Therefore, when finalizing the hyperparameters for the SO-INRD detection-aware adversary we restricted our search to a setting where the decrease in the success rate of the attack was at most $10\%$.

Since FO-INRD is based on sampling a random perturbation, we use another approach introduced by Carlini & Wagner (2017b) to minimize the expectation of the original loss function when averaged over the randomness used in the detection method. In particular, we estimate the expectation by computing the empirical mean of the loss over 50 samples from the same noise source. As for the case of SO-INRD we grid search over hyperparameters to achieve as low a TPR as possible while losing at most $10\%$ in the success rate of the attack. Table 3 shows the TPR in the adversary-aware setting with the best hyperparameters found for each method. The fact that SO-INRD still performs quite well in the adversary-aware setting is an indication that there is a fundamental trade-off between computing an adversarial example and minimizing $\mathcal{L}(s)$. This trade-off makes sense in light of Proposition 3.2, which shows that searching for an adversarial example in a small neighborhood will tend to increase $\mathcal{L}(s)$.

## 6 CONCLUSION

In this paper we introduce a novel algorithm INRD, the first method for identification of adversarial directions in the deep neural policy manifold. Our method is theoretically motivated by the fact that local optimization objectives corresponding to the construction of adversarial directions lead naturally to lower bounds on the curvature of the cost function $J(s, \tau)$. We have further shown empirically that the curvature of $J(s, \tau)$ is significantly larger at adversarial states than at clean observations, leading to a highly effective method SO-INRD for detecting adversarial directions in deep reinforcement learning. We additionally demonstrate that SO-INRD remains effective in the adversary-aware setting, and connect this fact to our original theoretical motivation. We believe that due to the strong empirical performance and solid theoretical motivation SO-INRD can be an important step towards producing robust deep reinforcement learning policies.

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
