# OpenReview forum: "Uncovering Directions of Instability via Quadratic Approximation of Deep Neural Loss in Reinforcement Learning"
_ICLR.cc/2023/Conference — Submitted to ICLR 2023_

### Official Review · Reviewer_FZra · 2022-10-22

**Confidence:** 2
**Correctness:** 4
**Technical Novelty And Significance:** 4
**Empirical Novelty And Significance:** 4
**Recommendation:** 8

**Clarity, Quality, Novelty And Reproducibility:**

The writing style of the paper is excellent and it does a good job in motivating central ideas which makes it easy for the reader to follow. Both the theoretical and empirical expositions seems to be of high quality. However, I am not familiar enough with related work to be able to judge whether the claims and results are sound. Source code is provided, demonstrating that the proposed method is very simple to implement, so I suppose the results can be easily reproduced.


**Strength And Weaknesses:**

Unfortunately, I am unfamiliar with the reasearch area of adversarial perturbations and thus can only give high level feedback with low confidence. I think the paper is a strong submission because:

- Detecting adversarial perturbations is important to make RL applicable to real-world scenarios.
- The paper is very well structured and written.
- The experimental section appears to be well fleshed out.
- The method is very simple to implement.


**Summary Of The Paper:**

The paper proposes a novel approach to detect adversarial state manipulations in deep Reinforcement Learning (RL). It provides a theoretical analysis that motivates that second-order gradient information w.r.t. a cost function contains information that allows to identifiy non-robust directions (SO-INRD) and, thus, adversarial perturabtions. Furthermore, it proposes a method to approximate the measure of curvature (i.e., the Hessian eigenvalues) in a computationally feasible manner. The method is validated empirically on DDQN agents trained on the Arcade Learning Environment, adversarially attacked by various different methods, and compared against several baseline methods for detecting adversarial perturbations.


**Summary Of The Review:**

The paper tackles an important problem, namely adversarial perturbations in RL, and proposes a theoretically well-founded and nicely motivated approach to detect these. The presentation is clear and seems to be theoretically and empirically sound. I propose to accept the submission, but me recommendation is with low confidence, so I might adjust my score after the discussion period.

---

> ### Author Response · Authors · 2022-11-18
> **Response**
>
> We would like to thank you for your positive comments, and we highly appreciate seeing acknowledgement of our contributions. Thank you again for investing your time to review our paper, and finding the topic of our paper important and our paper theoretically and empirically sound with theoretical and empirical expositions of high quality.

---

### Official Review · Reviewer_8ZPv · 2022-10-24

**Confidence:** 3
**Correctness:** 3
**Technical Novelty And Significance:** 3
**Empirical Novelty And Significance:** 2
**Recommendation:** 5

**Clarity, Quality, Novelty And Reproducibility:**

Clarity: The paper could be made somewhat more self-contained for those not deeply involved in the adversarial RL literature. For example:
- $\tau$ is not defined as the adversary's target explicitly.
- The paper discusses $J(s, \tau)$ and $J(s_0, \pi^*)$. Most of the motivation is about the former, but the algorithm is about the latter (because it does not assume that we know $\tau$). But there is not much support/intuition for why we can use $J(s_0, \pi^*)$ as a surrogate (likely because this is not a new insight of the paper).

Quality: see weaknesses. It is not clear how significant the paper's results are in the adversarial arms race.

Novelty: The authors' method builds on prior work, but the methods are sufficiently novel.

Reproducibility: The authors provide Python code in the appendix. I hope they will release their experimental code.

**Strength And Weaknesses:**

Strengths:
1. The paper suggested an interesting detection method for adversarial directions and conducted experiments under many different environments and attack methods.
2. The paper addresses important issues for making unstable reinforcement learning robust.

Weaknesses:
1. The authors claim implicitly that better detection of adversarial states leads to better performance (e.g., in average return). But they do not explicitly test this.
2. It is unclear why the particular test environments were selected, but they do not look randomly chosen.
3. This "adversarial defense" seems to suffer especially badly from the adversary being aware of it—they can now broaden their search to try to find states that avoid the curvature condition.

**Summary Of The Paper:**

This paper proposes a method of identifying adversarial directions for robustness in deep reinforcement learning. The authors show theoretically that there is a lower bound on the curvature of a cost function that is related to a local adversarial optimization. Empirically, they show that this curvature is greater at adversarial states. They develop a method based on this observation, which they show is more effective at identifying adversarial examples than some baselines.

**Summary Of The Review:**

The authors method performs impressively in the scenario where it is tested, which is a proxy for the objective the authors really care about. The authors do not justify why their experimental setup is the correct one for this problem. While the method has theoretical justification, the evaluation should be primarily empirical.

Post author response:
I don't understand the authors' response that it is enough to detect adversarial states. The purpose has to be to achieve a high reward in the presence of an adversary. If it is just adversarial manipulation detection, it seems like the problem isn't specific to RL.

---

> ### Author Response · Authors · 2022-11-10
> **Author Response**
>
> Thank you for allocating your time to provide feedback on our paper. Below we address your questions.
>
> 1. *” Average return”*
>
> You might have a confusion here. We do not claim that detection will yield a better average return. We claim that while there are currently problems with adversarial training, detection might help in identifying these non-robust directions with extremely high TPR scores.
>
> 2. “*Test Environments”*
>
> We tried to incorporate many games from the Arcade Learning Environment with complicated game semantics, perceptually distinct observations across MDPs, and variations in the action space. Note that the games used in our paper consist of the standard games used in adversarial deep reinforcement learning [1,2,3] with additionally more complicated MDPs from the Arcade Learning Environment.
>
> [1] Deep Reinforcement Learning Policies Learn Shared Adversarial Features Across MDPs. AAAI Conference on Artificial Intelligence, 2022.
>
> [2] Robust Deep Reinforcement Learning against Adversarial Perturbations on State Observations, NeurIPS 2020.
>
> [3] Stealthy and Efficient Adversarial Attacks against Deep Reinforcement Learning, AAAI 2020.
>
>
> 3. *”Adversarial arms race*”
>
> Recent work [1] demonstrates that the state-of-the-art adversarial training in deep reinforcement learning is vulnerable to quite standard adversarial attacks such as Carlini & Wagner and Elastic Net across states, across MDPs, and across algorithms. The fact that one of the major ways of achieving robust policies is still vulnerable to standard adversarial attack techniques makes our method that can identify these non-robust directions of significant interest towards protecting the policy stability under adversarial manipulations. In our paper, Section 4 in Table 1, Figure 2 and Figure 3 report results for both Carlini&Wagner and Elastic Net adversarial directions. The results reported in Table 1, Figure 2 and Figure 3 once more demonstrate that these adversarial directions can be identified with our proposed method, thus allowing the deep reinforcement learning policy to be more stable when these adversarial directions are introduced to its state observations.
>
>
>
> 4. *”Detection Aware”*
>
> Note that in the detection aware setting for the feature matching method SO-INRD performs more than 100 times higher than the prior methods in half of the MDPs, and more than 50 times better than the prior methods in the other half of the MDPs. Note that while the prior methods under detection aware adversaries experience performance decrease by 76%, our method only experiences a 9% decrease on average across MDPs.

---

### Official Review · Reviewer_9bNZ · 2022-10-25

**Confidence:** 3
**Correctness:** 2
**Technical Novelty And Significance:** 2
**Empirical Novelty And Significance:** 2
**Recommendation:** 5

**Clarity, Quality, Novelty And Reproducibility:**

The abstract and the beginning part of introduction might need some elaboration to clarify why reinforcement learning is studied and what is the difference to a standard classification problem. Otherwise, the claim "first one that focuses on detection of non-robust directions in
the deep reinforcement learning neural loss landscape" does not have full merit.

There are some inconsistent notations like $J(s, \pi)$ v.s. $J(s)$.

It is good to comment on the gap between theory and experiments. The theoretical results assume that a local minimum is obtained, while in experiments, this assumption is not always satisfied. Then what can we expect from the theoretical results?

**Strength And Weaknesses:**

=========== Strength ===========

Rather than directly working with the second order detection method, the paper starts with the first order method and indicates the inaccuracy of the first order Taylor approximation. This type of reasoning stimulates the interest of readers and provides background for later more involved discussions.

Apart from theoretical discussions, the paper provides abundant empirical results to showcase the effectiveness of the proposed method.

=========== Weakness ===========

The connection to reinforcement learning is pretty weak. Suppose we consider the policy learning as a classification problem, where the state $s$ is viewed as input feature and the policy output is the classification odds (on a simplex). Then the paper indeed discusses detecting adversarial directions in general classification problems. I am a bit confused why reinforcement learning problems are the main target here, providing that the paper does not consider any temporal dependencies.



**Summary Of The Paper:**

This paper proposes an algorithm for identification of adversarial directions in the deep neural policy manifold. The method is motivated from the insufficiency of first order approximation of the loss function; instead, the method relies on the second order information of the loss (curvature). Empirical results support the effectiveness of the proposed SO-INRD method.

**Summary Of The Review:**

My major concern is what is the differences of studying adversarial detection in RL compared to that in classification problems. I am giving an initial negative rating, and happy to involve in the discussion with the authors. If my concern is lifted, I will raise my score.

=========== Post author response ===========

Thank you for addressing my comments and questions!

I still believe temporal dependency should be comprehensively discussed and tackled by the methodology that aims at MDPs, instead of treating it as a future direction. Otherwise, the method is not very different from those in standard classification problems. Given the vast literature in robust classification, the paper's contribution can be undermined.

The gap between theory and experiments are not explained convincingly in response. I am dubious about whether the experiments support the theory, as assumption is violated.

---

> ### Author Response · Authors · 2022-11-10
> **Author Response**
>
> Thank you for dedicating your time to provide feedback on our paper. Below we respond to your questions.
>
> 1. *“Temporal dependence”*
>
> Thank you very much for raising this question. We see this question as a strength of our method that as a future research avenue that can be applied and extended to many different settings that do not carry temporal dependence. While our method achieves significant performances in identifying non-robust directions across MDPs that are independent from the temporal dependence of the state observations we believe as a future research objective it is also possible to optimize further to exploit the temporal dependence. This might be an interesting future research direction. However, one must keep in mind that as the identification method gets specifically tuned to a technique that exploits temporal dependence, it might also suffer more from the effects of the identification-aware adversary.
> Thus while the temporal independence of our method makes our technique more eligible to be easily adopted in many different settings, it also stands in the line of fundamental trade-off between temporal independence and the identification-aware adversary.
>
> 2. *“What can we expect from the theoretical results?”*
>
> While the theoretical results in Section 3 provide insights into the negative curvature of the deep neural policy at the unstable non-robust state observations, the empirical results from Section 4 to 5 demonstrate that the theoretical justifications and motivations into explaining the deep neural policy loss landscape around the adversarial observations indeed hold, and hence result in superior performance for the SO-INRD method.

---

> ### Author Response · Authors · 2022-11-18
> **Gentle Reminder**
>
> Dear Reviewer 9bNZ,
>
> Thank you again for providing feedback on our paper. The author response period will be ending today. Would there be a possibility for you to confirm that our response addressed your questions? If you have further questions, you can let us know in order to allow us to properly address them before the deadline.

---

> ### Comment · Area_Chair_XykW · 2022-11-24
> **Follow-up**
>
> Dear reviewer,
>
> I'm following up on this. It is important to have an accurate evaluation given scores of this paper. Please let me know if your decision has changed in any way.
>
> Thank you, Area Chair

---

### Official Review · Reviewer_Soiv · 2022-10-25

**Confidence:** 4
**Correctness:** 3
**Technical Novelty And Significance:** 2
**Empirical Novelty And Significance:** 2
**Recommendation:** 5

**Clarity, Quality, Novelty And Reproducibility:**

The paper is written well in my opinion. One comment I have about the writing is on Algorithm 1. It is not clear whether J is given to the algorithm (or estimated via e.g. Monte Carlo?). The authors should be more precise in their presentation and explicitly write J as part of the required inputs / explain how to estimate it. Otherwise, Algorithm 1 may be mis-interpreted.

**Strength And Weaknesses:**

Strengths.
-) The authors put forward a method to recover particularly sensitive states: states from which small perturbations can result in dramatic differences in the future. This is worthwhile from real-world perspective and can be a nice debugging tool for such applications.
-) The paper is well written in my opinion (besides of minor comments). The intuition is very clear.

Weaknesses.
-) The algorithm and method are quite simple and straightforward. On the other hand, the experiments are also quite basic (in the sense that the authors didn't show that their method leads to a new capability that was not achieved prior to this work).
-) The authors considered only detection. I believe it should be possible to built mitigations using a min-max approach (e.g., the one that was taken in the action robust framework https://arxiv.org/pdf/1901.09184.pdf). This would make this work more complete in my opinion.
-) There's a need to estimate the cost function using model free approaches. It might be the case that this due to errors at this step the algorithm would not detect the non robust states.



**Summary Of The Paper:**

The authors studied a method that can detect the presence of unstable states. They achieve that by measuring the gap between the perturbed cost and the first order approximation. I this gap is large (one the direction is chosen in a worst case fashion) then the second order curvature is large as well (in a worst case sense). The authors define such states as unstable.

Besides of suggesting this method, the authors also conducted experiments on the ALE environment.

**Summary Of The Review:**

The authors detect sensitive states by detecting gaps between first and second order approximation of the cumulative cost function. Assuming access to offline data, the authors can detect problematic states from which small perturbations can lead to different results.

---

> ### Author Response · Authors · 2022-11-10
> **Author Response**
>
> Thank you for allocating your time to provide feedback for our paper. Below we respond to your questions.
>
> 1. *“There's a need to estimate the cost function using model free approaches”*
>
> We believe you have some confusion here. We do not need to estimate the cost function. The cost function belongs to the reinforcement learning agent; and hence intrinsically is a  part of the detection mechanism. The cost function does not need to be estimated. Thank you for indicating this confusion.
>
> 2. *”Action robust framework”*
>
> Our paper focuses on non-robustness in state observations, while the paper you refer to [1] is for action robustness. To be more specific, the correct analogy would be studies working on adversarial robustness in state observations such as [2] which is a more recent study that has been accepted as a spotlight presentation. However, these state-of-the-art adversarial training methods quite recently have been shown to be vulnerable towards very standard adversarial attack techniques such as Carlini & Wagner and Elastic Net across state observations, across MDPs, and across algorithms [3]. Thus the fact that certified robust adversarial training techniques are still experiencing non-robustness under standard adversarial directions makes our identification method of greater interest to the community.
>
> [1] Chen Tessler, Yonathan Efroni and Shie Mannor. Action Robust Reinforcement Learning and Applications in Continuous Control, ICML 2019.
>
> [2] Robust Deep Reinforcement Learning against Adversarial Perturbations on State Observations, NeurIPS 2020. [Spotlight Presentation]
>
> [3] Deep Reinforcement Learning Policies Learn Shared Adversarial Features Across MDPs. AAAI Conference on Artificial Intelligence, 2022.
>
>
> 3. *”Detection”*
>
> Our paper is the first one that focuses on detecting adversarial directions introduced to state observations in deep reinforcement learning. Can you please elaborate what you mean by “ to a new capability that was not achieved prior to this work”?

---

> > ### Comment · Reviewer_Soiv · 2022-11-18
> > **Thanks for your comment.**
> >
> > 1. Don't you assume the cost function is being given to the learning agent?
> > 2.  I agree there's a big difference. I eluded to the fact that the min-max approaches might be able to offer a mitigation for the non-robustness detected in the system by your approach. Wouldn't you agree that a min-max approach is natural step to take towards robustification of the system?
> > 3. First, as far as I understand, the suggested approach allows to detect problematic states (by considering the sensitivity of the cumulative cost). Conceptually, such approach makes much sense. I believe that improving the experiments and showing this approach can work in larger scale (i.e., considering challenging/real-world domains) would make this work more complete. I do want to emphasize that is only my subjective opinion.
> >
> > To summarize, I believe that either offering mitigations (e.g., using min-max approach) or scaling the experiments are needed next steps to make in my opinion.

---

> > > ### Author Response · Authors · 2022-11-18
> > > **Thank you for your response.**
> > >
> > > 1. *"Don't you assume the cost function is being given to the learning agent?"*
> > >
> > >
> > >  The cost function of the agent intrinsically belongs to the agent itself, and it might cause confusion to refer to “given” in this context. The reason for this is there are two entities in this setting: **the agent** and **the adversary**. To understand this better let us look at the detection aware adversary setting. In this setting it is suitable to refer to that the detection system information as **given** to the adversary, because the detection system information intrinsically belongs to the agent; therefore, for this setting it is considered that this information is given (i.e. provided) to the adversary. However, the cost function of the agent already intrinsically belongs to the agent. Thus, it should not be referred to as given.
> > >
> > > 2. *"I agree there's a big difference. I eluded to the fact that the min-max approaches might be able to offer a mitigation for the non-robustness detected in the system by your approach. Wouldn't you agree that a min-max approach is natural step to take towards robustification of the system?"*
> > >
> > > The adversarial training method SA-DQN [9] is itself a min-max approach to robustification. The algorithm is based on minimizing the maximum possible policy change under norm-bounded perturbations to the state observations. In particular, the maximum possible policy change is estimated via a convex relaxation upper bounding the value of the neural policy within an $\ell_p$-norm bounded ball at each state encountered in training. Thus, as we have also explained in the previous response, these certified adversarial training methods (e.g. SA-DQN) have already been shown to be vulnerable to standard adversarial attacks in recent studies [12]. Thus, at this point given all this prior work we are not entirely sure if the min-max approach will lead towards complete robustification of the system.
> > >
> > >
> > > 3. *"First, as far as I understand, the suggested approach allows to detect problematic states (by considering the sensitivity of the cumulative cost). Conceptually, such approach makes much sense. I believe that improving the experiments and showing this approach can work in larger scale (i.e., considering challenging/real-world domains) would make this work more complete. I do want to emphasize that is only my subjective opinion."*
> > >
> > > Thank you for referring to this item as your subjective opinion. We also agree that utilizing our detection algorithm for real life problems is an interesting direction for future research avenues. However, it is important to note that the Arcade Learning Environment is the **main baseline** for deep reinforcement learning algorithm development [1,2,3,4,5,6, 7,8] and deep reinforcement learning robustness investigation [9,10,11,12]. Thus, in the reinforcement learning community it has been the main benchmark that allows researchers to be able to provide **transparent** and **consistent** comparisons.
> > >
> > > [1] Human-level control through deep reinforcement learning, Nature 2015.
> > >
> > > [2] Deep Reinforcement Learning with Double Q-learning, AAAI 2016.
> > >
> > > [3] Dueling Network Architectures for Deep Reinforcement Learning, ICML 2016.
> > >
> > > [4] Implicit Quantile Networks for Distributional Reinforcement Learning, ICML 2018.
> > >
> > > [5] Bootstrapped Metal-Learning, ICLR 2022.
> > >
> > > [6] Muesli: Combining Improvements in Policy Optimization, ICML 2021.
> > >
> > > [7] When to use parametric models in reinforcement learning?, NeurIPS 2019.
> > >
> > > [8] Policy Improvement by Planning with Gumbel, ICLR 2022.
> > >
> > > [9] Robust Deep Reinforcement Learning against Adversarial Perturbations on State Observations, NeurIPS 2020. [Spotlight Presentation]
> > >
> > > [10] Adversarial attacks on neural network policies, ICLR 2017.
> > >
> > > [11] Stealthy and Efficient Adversarial Attacks against Deep Reinforcement Learning, AAAI 2020.
> > >
> > > [12] Deep Reinforcement Learning Policies Learn Shared Adversarial Features Across MDPs. AAAI 2022.

---

> ### Author Response · Authors · 2022-11-17
> **Gentle Reminder**
>
> Dear Reviewer,
>
> The author response period will be finalized on 18th of November. Would it be possible for you to confirm that your questions have been addressed? If you have any further questions, we also would be happy to discuss.

---

### Decision · Program_Chairs · 2023-01-20

**Decision:**

Reject

**Justification For Why Not Higher Score:**

Three out of four reviewers do not believe that the paper should be accepted. The reviewer with the score of 8 is low-confident and decided not to champion the paper.

**Justification For Why Not Lower Score:**

N/A

**Metareview: Summary, Strengths And Weaknesses:**

The paper suggests two methods to detect adversarial attacks on the state of an RL agent. Two test statistics are defined based on the first and second order Taylor series approximation of a loss function. It is argued that the value of these test statistics are larger at the points perturbed by an adversary compared to unperturbed ones.

Although reviewers find the detection method interesting, the majority of them do not believe that the paper is ready to be accepted. Two major concerns are:

1) The paper focuses on the detection of adversarial attacks, but it should also show the effect on the performance of the agent (Reviewers 8ZPv and Soiv). This requires a procedure that uses the detection method to decide what to do at the presence of an adversary, so that it acts more robustly against an adversary. The reviewers believe that the paper is incomplete without that.

2) It is not clear why the detection method is studied within the context of RL when the method does not really benefit from the fact that an RL problem is being solved (Reviewer 9bNZ). The methods look generic enough that they could be tested and compared on classification problems. Although this might be considered a strength of the methods, it also brings up the question of why the methods are not empirically studied and compared with other detection methods in the classification literature, which is much larger than the adversarial attacks in the RL context.

At the end of the day, the negative reviewers were not convinced by the authors' rebuttal. The positive reviewer did not defend the paper in the private discussion as they are not an expert in the area. As a result, unfortunately I cannot recommend acceptance of this work. I encourage the authors to consider the feedback in order to improve their paper, as the reviewers believe that there is some merit in this work.